# Unveiling the Genetic Secrets of Chinese Indigenous Pigs from Guizhou Province: Diversity, Evolution and Candidate Genes Affecting Pig Coat Color

**DOI:** 10.3390/ani14050699

**Published:** 2024-02-23

**Authors:** Ziping Hu, Yanfang Su, Wencheng Zong, Naiqi Niu, Runze Zhao, Ruiping Liang, Lixian Wang, Yiyu Zhang, Longchao Zhang

**Affiliations:** 1State Key Laboratory of Animal Biotech Breeding, Institute of Animal Sciences, Chinese Academy of Agricultural Sciences (CAAS), Beijing 100193, China; hu07232021@163.com (Z.H.); 15103875909@163.com (Y.S.); zongzone@outlook.com (W.Z.); 18734450897@163.com (N.N.); zhaorunze08@163.com (R.Z.); 2College of Animal Science and Technology, Qingdao Agricultural University, Qingdao 266109, China; iaswlx@263.net; 3Beijing Changping District Center for Animal Disease Prevention and Control, Beijing 102200, China; ruiping80@126.com; 4Key Laboratory of Animal Genetics, Breeding and Reproduction in the Plateau Mountainous Region, Ministry of Education, College of Animal Science, Guizhou University, Guiyang 550025, China

**Keywords:** population structure, conservation, genetic variability, population relationship, coat color

## Abstract

**Simple Summary:**

There are few reports on local pig breeds in Guizhou province. In this work, the population structure, genetic diversity, and selection characteristics of three foreign pig breeds and seven Guizhou local pig breeds were studied. Principal component analysis, adjacent tree analysis, and ADMIXTURE analysis showed that Chinese and Western pig breeds had different ancestral sources. Jianhe White Xiang pig (JHBX) had a distant genetic relationship with the other six local pig breeds in Guizhou province and had unique genetic characteristics. The genetic diversity of the JHBX population was low, and the genetic diversity of the remaining six Chinese native pig populations was generally at a moderate level. In addition, we also screened the candidate genes that affect the coat color phenotype of pigs in Guizhou province through selection signals. Our findings will advance our understanding of the genetic mechanisms underlying these germplasm characteristics.

**Abstract:**

The local pig breeds in Guizhou possess exceptional meat quality, robust adaptability, and resilience to harsh feeding conditions, making them ideal for producing high-quality pork. With over 10 local pig breeds in the region, we focused on 7 specific breeds: Baixi pigs (BX), Congjiang Xiang pigs (CJX), Guanling pigs (GL), Jianhe White Xiang pigs (JHBX), Jiangkou Luobo pigs (JKLB), Kele pigs (KL), and Qiandong Hua pigs (QDH). Unfortunately, these breeds face threats such as introduced species and inbreeding, resulting in a decline in population size and numbers. To better protect and utilize these breeds, we employed genome-wide single-nucleotide polymorphism (SNP) markers to investigate the population structure, genetic diversity, and selection characteristics of 283 pigs across these seven breeds. Our findings revealed distinct ancestral sources between Chinese and Western pig breeds, as demonstrated by principal component analysis, adjacent tree analysis, and ADMIXTURE analysis. Notably, JHBX exhibited a distant genetic relationship from the other six local pig breeds in Guizhou province, showcasing unique genetic characteristics. While the genetic diversity of the six Chinese native pig populations, excluding JHBX, was generally moderate in Guizhou province, the JHBX population displayed low genetic diversity. Therefore, it is imperative to intensify selection efforts to prevent inbreeding decline in JHBX while further enhancing the protection measures for the other six pig populations. Additionally, we identified candidate genes influencing the size disparity among pigs in Guizhou province through signal selection. Our study outcomes serve as a reference for developing effective conservation and utilization plans for pig breeds in Guizhou province and deepen our understanding of the genetic mechanisms underlying pig body size.

## 1. Introduction

China has one of the richest genetic resources for local pig breeds worldwide, accounting for approximately one-third of the global breeds [1]. It has a vast territory and a wide distribution of local pig breeds, forming many excellent pig breeds. For example, Meishan pigs [2] and Erhualian pigs [3] in East China have large litter sizes, and Erhualian pig breed is one of the famous high-yielding pig breeds in China; Luchuan pigs and Bama pigs in South China have thin skin and excellent heat resistance [4,5]; Min pigs [6] and Tibetan pigs [7] have strong cold tolerance; and there are varieties resistant to rough feeding, with strong adaptability and excellent meat quality. These pig breeds play a key role in the Chinese pig industry and contribute to the development of international commercial breeds.

Western commercial pigs have gradually dominated the Chinese pig industry in the past few decades owing to their fast growth rate, high meat yield, and low feed/meat ratio [8,9]. Moreover, the litter size of Western pig breeds (e.g., Landrace from Denmark and Large White from France) exceeds that of most local pigs in China now. This is one of the important reasons why Western pig breeds completely dominate the market in China. The large number of Wester pig breeds introduced and unplanned crossbreeding methods have posed a great threat to China’s indigenous pig breeds, especially after the outbreak of African swine fever. The population of native Chinese pig breeds (including those in the Guizhou region) has sharply declined, threatening the germplasm of Chinese pigs. Therefore, it is important to study the population structure and genetic diversity of native Chinese pig breeds using genomic methods to maintain genetic diversity, avoid inbreeding, and formulate effective conservation strategies.

Guizhou province is in southwest China on the Yunnan–Guizhou Plateau, a subtropical humid monsoon climate area rich in pig varieties. A number of pig breeds were bred owing to their unique geographical conditions, including Baixi pigs (BX), Guanling pigs (GL), Kele pigs (KL), and Jianhe White Xiang pigs (JHBX). These breeds provide valuable genetic resources for scientific research and further genetic improvements in the pig industry. However, the genetic diversity of these varieties remains unclear, and it has not been reported previously. Therefore, 283 individuals from seven Guizhou provincial pig breeds and three European breeds were selected to reveal their genetic diversity, population structure, and historical mixing characteristics. This will be conducive to the effective conservation and utilization of these seven varieties in Guizhou and will promote our understanding of the genetic mechanisms underlying their germplasm characteristics.

## 2. Materials and Methods

### 2.1. Animals and SNP Genotyping

A total of 283 pigs from 10 breeds were investigated, comprising 7 breeds from China and 3 breeds from Europe. Of those, 193 were from seven breeds in Guizhou province, China (BX, Congjiang pigs, GL, Jianhe white pigs, Jiangkou Radish pigs, KL, and Qiandong Hua pigs), while the remaining 90 pigs were Duroc, Landrace, and Great White European breeds.

Genomic DNA was extracted from ear samples using conventional phenol–chloroform methods and diluted to a final concentration of 50 ng/μL. Genome-wide SNPs were detected using the Porcine SNP50 BeadChip (Illumina, San Diego, CA, USA). In total, 50,697 SNPs were identified for subsequent analyses. The genotype files were converted to plink input files. Plink (v1.90) was used for the quality control of SNP data, and the criteria were as follows: Individuals with an individual deletion rate > 10%, individuals with an SNP deletion rate > 10%, individuals with a minor allele frequency < 5%, and individuals with sex chromosome deletion were excluded. At the same time, the extracted data were combined with previously existing Duroc, Landrace, and Large White pig 50 K SNP chip data, and a follow-up analysis was performed.

### 2.2. Estimation of Population Genetic Differentiation

Genome-wide Complex Trait Analysis (GCTA) (v1.93) software was used to calculate the average proportion of shared alleles (Dst), and the genetic distance between all pairs of individuals was calculated as (1-Dst). Using the Bionj function in the R package, an adjacency tree was constructed based on the (1-Dst) matrix, and the iTOL website (https://itol.embl.de/ (accessed on 29 June 2023)) was used to identify the constructed adjacency tree.

### 2.3. Genetic Differentiation Index between Breeds

The genetic differentiation index (*Fst*) between the two breeds was calculated using vcftools. The *Fst* value is determined as follows: the low is 0–0.05, the medium is 0.05–0.15, the high is 0.15–0.25, and the high genetic differentiation is above 0.25 [10].
$$Fst=\frac{\left(Ht-Hs\right)}{Ht}$$
where *Fst* is the genetic differentiation index; *Ht* is the mean heterozygosity in subpopulations; and *Hs* is the mean degree of heterozygosity in a composite population.

### 2.4. Principal Component Analysis

Principal component analysis (PCA) was performed using GCTA to determine the relationships among these varieties. First, we calculated the ten principal components of these varieties. The first two principal components were selected and visualized using R.

### 2.5. Inferring Population Structure and ADMIXTURE

Population isolation usually prevents free migration and mating between subgroups. We used the default parameters of the ADMIXTURE (v1.3.0) software to estimate the consanguinity structure of the filtered data, with the number of ancestor populations K ranging from 2 to 20, and an R script was used to visualize consanguinity structure mixing.

TREEMIX (v1.13) was used to further investigate the history of interspecific gene flow in pigs [11]. This program runs on a VCF file containing the studied pig breeds, with Duroc, Landrace, and Great White pigs selected as outpopulations. Initially, we calculated allele frequencies using VCF files and converted the files to TREEMIX format. We set the parameter to -k 1000 since the TREEMIX runtime assumes that SNPs are unchained to prevent starting the TREEMIX runtime with SD = 0. TREEMIX runs with m values between 1 and 10, with five cycles per m value. The OptM program was used to determine the optimal number of migrated edges (m) [12].

### 2.6. Analysis of Genetic Diversity Indices

To compare the genetic diversity of different pig populations, we calculated the expected heterozygosity (*He*) for each of these breeds using plinkV1.9, as well as the observed heterozygosity (*Ho*), minor allele frequency (*Maf*), and inbreeding coefficient (*Fis*). The *Fis* for each variety was calculated based on the observed homozygous genotype and the expected number using the following formula [13,14]:$$Fis=\frac{O\left(HOM\right)-E(HOM)}{N\left(NM\right)-E(HOM)}$$
where *Fis* is the inbreeding coefficient; *O*(*HOM*) is the observed number of homozygotes; *E*(*HOM*) is the expected number of homozygotes; and *N*(*NM*) is the number of non-missing genotypes.

### 2.7. Runs of Homozygosity Analysis

Plink (v1.90) software was used to detect the homozygous regions of a certain number and density of SNPs in the genome, namely runs of homozygosity (ROHs). The main parameters were as follows: The sliding window was set as 20 SNPs along the chromosome, the number of heterozygotes in each sliding window did not exceed one, the minimum length was >1000 kb, the maximum distance between two SNPs was 100 kb, the number of SNPs/ROH was >30, and the minimum density of SNPs was >50 kb/SNP [15,16]. R (v4.2) was used to calculate the inbreeding coefficient (*F*_ROH_) based on ROHs, and the homozygosity measured from genomic data based on ROHs (*F*_ROH_) was defined as the total length of the genome covered by an ROH divided by the total length of the genome covered by the SNP or sequence [17], as shown below:$$F_{\mathrm{ROH}}=\frac{\sum L_{ROH}}{L_{AUTO}}$$
where ∑*L_ROH_* is the sum of the total length of all ROHs detected in an individual, and *L_AUTO_* is the total length of the autosomal genome.

### 2.8. Selective Scan Analysis

Guizhou local pig breeds have a variety of coat color phenotypes. JHBX’s coat color is two-end black; KL’s coat color is mostly black, but a few are brown; and the coat color of JKLB and CJX is black. In order to screen the genes affecting coat color phenotypes, we performed the selective scanning of JHBX and KL with CJX and JKLB, who have black coat color, respectively. We used the following strategies to conduct the genome scanning of pig breeds in Guizhou province. We used nucleotide diversity (θπ) to detect the selection characteristics of local pig breeds in Guizhou province. By using vcftools, nucleotide diversity was estimated using a 200 kb window and a 10 kb step size. The regions with significantly high π values (the first 1% of π values covered) were selected as candidate regions for genomic selection. R software was used to visualize the π values at the whole genome level. Referring to the pig reference genome (Sscrofa 11.1) on the Ensembl website (https://asia.ensembl.org/index.html, accessed on 27 October 2023), candidate genes were selected within a 100 KB range upstream and downstream of the significant region. In order to better understand the gene functions and signaling pathways of the identified candidate genes, GO and KEGG pathway enrichment analyses were performed using kobas. A *p* value of less than 0.05 indicated a significant enrichment pathway.

## 3. Results

### 3.1. Phylogenetic Relationships and Genetic Differentiation

To compare the genome-wide genetic relationships and degrees of differentiation between local pig breeds in Guizhou province and European pig breeds, initially, we constructed adjacency neighbor-joining (NJ) trees based on the identity-by-state (IBS) distance matrix for all 283 pigs of the 10 Eurasian breeds (Figure 1B). In general, Chinese local pig breeds and Western commercial breeds were clustered on two sides of the evolutionary tree. Almost all breeds were independently branched, and individuals of the same breed were clustered together; BX had one individual clustered on the branch of CJX, except KL. None of the KL individuals clustered into a single branch, although they were dispersed among other Guizhou and European pig breeds.

ADMIXTREE analyses were performed for the 10 Eurasian varieties, with K values ranging from 2 to 10 and 16, to assess their evolutionary origins in different populations in Guizhou province (Figure 1E). Chinese and Western pig breeds appeared as two distinct clusters with different ancestral lineages when K = 2, as previously reported [9,18]. However, the genetic background of some Guizhou pigs was mixed with that of a small number of Western pig breeds. When K = 3, JHBX was separated from the other local varieties in Guizhou province, indicating the unique genetic characteristics of JHBX. When the K value was 6, BX was differentiated from the local pig breeds in Guizhou province. Followed by K = 7, CJX and QDH had the same differentiation cluster, and JKLB, GL, and KL were grouped in another differentiation cluster; all breeds had different differentiation clusters when K = 10. The lowest cross-validation error was observed for K = 16. At the same time, KL pigs had the most mixed bloodlines, which was consistent with the results of the NJ trees, and KL pigs did not cluster into independent branches (Figure 1E).

Principal component analysis was performed on local pig breeds in Guizhou province and European pig breeds. The two largest principal components, PC1 and PC2, were used to map the results (Figure 1C,D). The Chinese breeds were clearly distinguished from the European breeds according to PC1, as shown in Figure 1C. Meanwhile, PC2 showed genetic differentiation between the DRC, LR, and LW breeds. PC1 and PC2 reflected the differences among the seven local varieties in Guizhou province when European varieties were removed from the dataset (Figure 1D). JHBX was distant from the other six varieties. This further indicated the unique genetic characteristics of JHBX. CJX was found to be closely related to QDH, and KL was found to be closely related to BX. The distribution pattern of one BX individual was near the CJX population, and the remainder clustered together. This agrees with the results of the NJ and ADMIXTURE analyses. JHBX exhibited the longest branch in the NJ tree. According to the results of the ADMIXTURE analysis, JHBX had different differentiation clusters from the other pig breeds in Guizhou province when K = 3.

We subsequently performed a TREEMIX analysis to identify the migration events that occurred in all 10 species. The OptM package was used to select the optimal number of migration edges (m value), and the optimal m value was 2. Therefore, we set the migration event to 2. The migration events that occurred in all 10 species were determined. Among the 10 varieties, GL showed the infiltration characteristics of DRC, and JKLB showed the infiltration characteristics of CJX (Figure 2). These migration events are consistent with the ADMIXTURE population structure.

### 3.2. Genetic Diversity Index

Five genetic diversity indices were calculated to compare the genetic variability of these ten varieties: *Ho, He, MAF*, *F*_ROH_, and *FIS*. Table 1 summarizes the genetic diversity parameters of the varieties. Across all populations, the expected heterozygosity ranged from 0.1342 (JHBX) to 0.3458 (KL), whereas the observed heterozygosity ranged from 0.1501 (JHBX) to 0.3914 (JKLB). *Ho* and *He* were higher in European pig breeds than in most Guizhou pig breeds, and *Ho* was higher than He in all pig breeds. The inbreeding coefficients were negative for all varieties, ranging from −0.26 (JKLB) to −0.05 (QDH) (Figure 3).

The ROH length and frequency can reflect population history [19]. The longer the ROH is, the closer the relatives are, and the longer the ROH fragments are, the higher the possibility of inbreeding in the family is [15]. *F*_ROH_ in these groups ranged from 0.008 (KL) to 0.038 (JHBX).

The *Fst* was used to estimate the values of all autosomal information markers at a genome-wide level to determine the degree of differentiation between Chinese and Western pig populations. The values for all pairwise population comparisons of the *Fst* are shown in Table 2. DRC exhibited the highest *Fst* with JHBX compared to other populations, and it also showed the greatest difference from JHBX. The populations of Chinese pigs and European pigs were found to be very different, with *Fst* values greater than 0.25. The *Fst* values of local pig breeds in Guizhou province significantly varied, from the smallest *Fst* value of 0.10 to the largest *Fst* value of 0.41.

### 3.3. Screening of Candidate Genes Affecting Pig Coat Color

We used nucleotide diversity analysis (θπ) to detect the candidate genes affecting pig coat color phenotypes. We established two comparison models. First, nucleotide diversity analysis was carried out based on JHBX with CJX, and JKLB. The second model was based on nucleotide diversity analysis of the presence of the brown coat phenotypes of KL with CJX, and JKLB.

Firstly, based on the analysis of JHBX, 1566 significant intervals were detected in JHBX, 796 genes were identified within these significant intervals, and these candidate genes were analyzed with the KEGG pathway enrichment analysis using kobas. The results of the enrichment analysis showed that 284 KEGG signaling pathways were enriched, and of those, 65 pathways were significantly enriched (*p* < 0.05). Among them, four signaling pathways were enriched, namely the PI3K/Akt signaling pathway, the MAPK signaling pathway, the Wnt signaling pathway, and the melanoma signaling pathway, which may be involved in the process of animal coat color generation. Notably, 43 genes were identified in these four pathways, including *PDGFRB*, *RB1*, *FGF7*, *BAD*, *DDB2*, *RAF1*, etc.

Based on the KL analysis results, 2164 significant regions were detected in KL, and 825 genes were identified within these significant regions, which were analyzed using the KEGG pathway enrichment analysis. The results showed that 284 KEGG signaling pathways were enriched, and 95 of these pathways were significantly enriched (*p* < 0.05). Three signaling pathways were enriched, namely the MAPK signaling pathway, the melanin generation pathway, and the PI3K/Akt signaling pathway, which may be involved in the process of animal coat color production. We identified 35 genes in these three signaling pathways, including *KIT*, *MAPK1*, *KRAS*, *PRKCG*, etc.

## 4. Discussion

### 4.1. Historical Relatedness and ADMIXTURE Analysis

The historical correlation and mixability of seven Chinese breeds in Guizhou and three European pig breeds were revealed through the analysis of adjacent trees, PCA, ADMIXTURE, and TREEMIX results.

The local pig breeds in Guizhou province and the European pig breeds were located in two large branches of the evolutionary tree in the NJ tree constructed based on IBS values. Furthermore, the PCA diagram proved that there was a significant difference between the Chinese and European pig breeds. ADMIXTURE ancestry composition analysis showed that the Chinese and Western pig breeds had two differentiated clusters when K = 2, indicating different ancestral lineages. This is consistent with previous reports indicating that European and Chinese pig breeds have different ancestral origins [8,20]. Wang et al. [8] used a 60 K SNP chip to study three pig breeds in Jiangxi province and found that Chinese and European pigs were clustered into two independent branches, whereas the Chinese–European hybrid breed (Sutai) was located in the middle between the European and Chinese branches in the NJ tree; the obvious difference between Chinese and European pig breeds was also reflected in the PCA diagram. Chen et al. [4] also found significant genetic differences between Chinese and European domestic pigs based on the population genetic analysis of 266 Eurasian wild pigs and domestic pigs. TREEMIX analysis showed that KL had the infiltration characteristics of DRC, which was of certain confidence because some KL had a brown coat color, consistent with the brown coat color phenotype of DRC. There was a migration event from CJX to JKLB, possibly owing to their proximity and the possibility of gene exchange.

### 4.2. Genetic Diversity

Chinese native pigs have richer genetic variability than European commercial pigs [18,21]. Therefore, they are expected to have higher Ho values than European commercial pigs. However, Ho levels in the three European varieties were higher than those in CJX, GL, JHBX, and QDH. The reason for the inconsistency with our expectations may be the small effective population size of the current sample and the underrepresentation of the sample. The degree of heterozygosity is an indicator of genetic variation in a population. Inbreeding may occur in the population when the observed heterozygosity is lower than the expected heterozygosity. The population may have historically differentiated when the observed heterozygosity is higher than the expected heterozygosity. The observed heterozygosity of all the tested sample groups was higher than their expected heterozygosity. This may indicate that the sample groups were historically differentiated. Fis represents the inbreeding coefficient of an individual relative to a subpopulation, that is, the average inbreeding coefficient of a subpopulation. Lower Fis values (including negative values) correlate with higher heterozygosity; higher Fis values correlate with higher homozygosity. In this study, the Fis values of all populations were negative. This indicated that the degree of inbreeding in these populations was low. The length and frequency of the ROH reflect the history of population kinship, and the number of long ROH fragments is positively proportional to the possibility of inbreeding within a family. Higher numbers correlate with greater possibilities of inbreeding [22]. In this study, the *F*_ROH_ in these populations ranged from 0.008 (KL) to 0.038 (JHBX) in the inbreeding coefficients calculated based on the ROH. The results of previous studies indicate that the average inbreeding coefficient is 0.11 and 0.026 in a Licha black pig population [23] and a Liangshan pig population based on the ROH [24]. The degree of inbreeding in the JHBX population in this study was higher than in the other breeds, but overall, all breeds were less inbred. In addition, the JHBX breed had a larger *F*_ROH_ than the modern European breeds and had the most ROHs, perhaps due to inbreeding in their ancestral populations, leading to a reduction in genetic diversity, which was reflected in the breed’s much lower Ho values. In contrast, the shorter ROHs and the lower degree of LD in KL pigs reflect the richness of their genetic variability. This suggests that the JHBX population needs further selection efforts to prevent inbreeding decline.

### 4.3. Coat Color Candidate Gene Screening

Coat color is one of the important characteristics of animal breeds. The head and tail of JHBX are black, and the middle is white, and some KL breeds have brown coat phenotypes. Based on the difference in coat color, the whole-genome nucleotide scanning analysis was performed on the pigs with black coat color phenotypes, and four and three related pathways were found to be involved in coat color formation for JHBX and KL, respectively. Among them, the kinases MEK and ERK in the MAPK signaling pathway are involved in the activation of melanocyte receptors, and the ligands activate the complex mechanism (Ras-Raf-MEK-ERK) by binding to the extracellular domain of the receptor, resulting in the upregulation of *MITF* [25]. Another underlying mechanism of the activation of the MAPK pathway is the interaction of endothelin (*EDN*) with its receptor. Zhou et al. reported that Selaginellin (SEL) inhibited the mitogen-activated protein kinase (MAPK) signaling pathway. Then, the expression of microphthalmia-associated transcription factor *(MITF*) and the downstream genes tyrosinase (*TYR*) and tyrosinase-associated protein 2 (*TYRP2*) were downregulated to inhibit melanin production [26]. Interleukin-10 (IL-10) can activate the PI3K/Akt and JAK/Stat3 pathways, and the former leads to the activation of the classical NF-κB pathway and the inactivation of GSK-3β, further upregulating melanin production [27]. Wnts is a cysteine-rich secreted glycoprotein, and the Wnt signaling pathway is involved in cell migration, proliferation, differentiation, and the self-renewal of stem cells [28], including neural crista-derived melanocyte development and migration [29]. Among the selected candidate genes, SUN et al. used CRISPR/Cas9 technology to create two mouse models, *KIT* D17/+ to simulate splicing mutations, and *KIT* Dup/+ to partially simulate *KIT* gene repeat mutations in dominant white pigs. The combination of these two mutations was found to reduce the phosphorylation of proteins associated with the PI3K and MAPK pathways, which may be related to the impaired melanoblast migration observed during embryonic development, ultimately leading to a dominant white phenotype [30]. In alpaca melanocytes, *IGF1* has been shown to improve melanin production via the cyclic AMP (cAMP) pathway [31]. Infarinato et al. [32] found that after the WNT-mediated activation of melanocyte stem cells, BMP and WNT pathways synergically triggered the occurrence of proliferative progeny by promoting *LEF1*- and *MITF*-dependent differentiation. Sastry et al. showed that ERK1/2, AKT, PKA, and PKC are important kinases responsible for melanocyte cell protection. Each of these pathways/kinases is partially responsible for survival, and they strictly regulate survival by inducing *BAD* phosphorylation. Melanocyte survival is mediated by a complex network of Bad-dependent and Bad-independent pathways [33]. The candidate genes of the above pathways can be used as candidate genes affecting coat color.

## 5. Conclusions

Our study revealed the genetic diversity, historical correlations, and population structure of seven pig breeds in Guizhou in a more comprehensive manner. In general, the genetic diversity of the six Chinese native pig populations (except for JHBX in Guizhou province) was at a medium level, while the genetic diversity of the JHBX population was low. Further selection analysis should be performed to strengthen the efforts toward preventing inbreeding decline; however, the protective effects on the other six pig populations should be further improved. At the same time, we identified a new set of candidate genes that may have an impact on phenotypic differences in coat color in pigs. In summary, we conducted a comprehensive genome-wide survey of seven native Chinese pig populations in Guizhou province. We believe that the results of this study will provide a good basis to develop national programs for the conservation and utilization of these pig populations. Furthermore, our findings will advance our understanding of the genetic mechanisms underlying the germplasm characteristics of these breeds.

## Figures and Tables

**Figure 1 animals-14-00699-f001:**
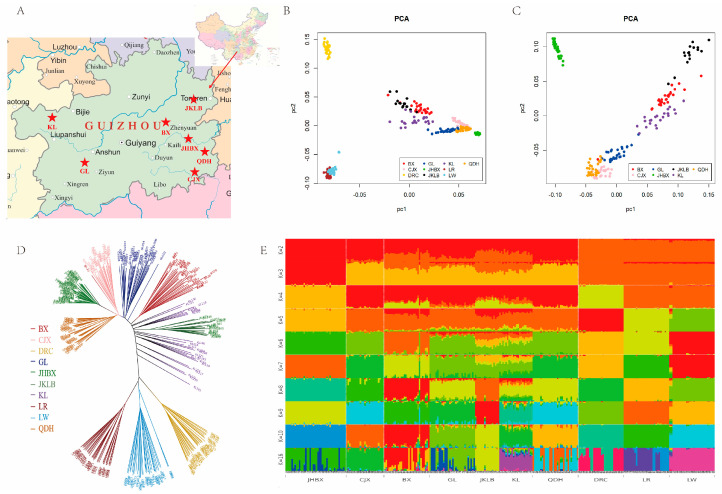
Phylogeny and population structure of 283 pigs from 10 Eurasian breeds: (**A**) The geographical location of seven indigenous pig breeds in Guizhou province. The seven breeds included Baixi (BX), Congjiang Xiang (CJX), Guanling (GL), Jianhe White Xiang (JHBX), Jiangkou Luobo (JKLB), Kele (KL), and Qiandong Hua (QDH). (**B**) Neighbor-joining tree of 283 pigs from the 10 breeds, including the 7 Guizhou breeds mentioned above, and Duroc (DRC), Landrace (LR), and Large White (LW). (**C**) Principal component (PC) plots of 283 pigs from the 10 pig breeds. The first (PC1) and second component (PC2) are shown, and the percentage represents the proportion of the corresponding principal component. (**D**) Principal component (PC) plots of 193 pigs from the 7 pig breeds in Guizhou province. (**E**) The ancestry of the 10 breeds was analyzed by ADMIXTURE with the assumed number of ancestries (K) from 2 to 10 and 16. Each color represents one ancestral cluster. All breeds are separated by dotted lines.

**Figure 2 animals-14-00699-f002:**
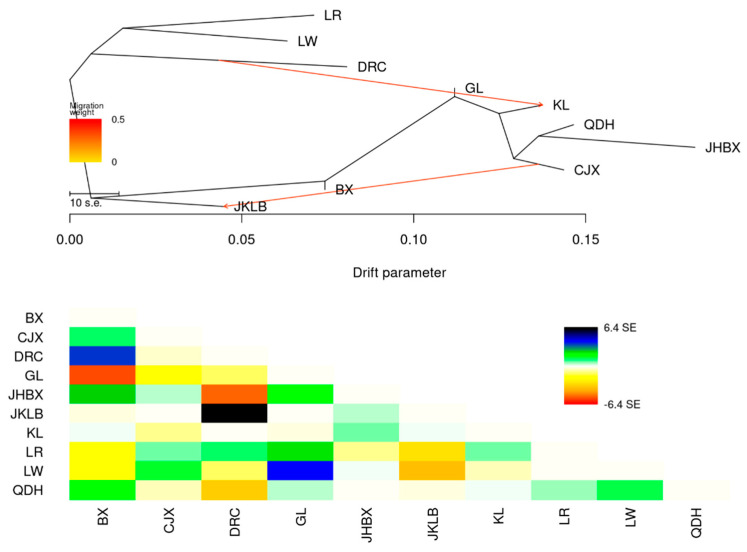
Population splits and ADMIXTURE analysis of all 10 Eurasian pig breeds were analyzed using TREEMIX. Arrows indicate migration events. A spectrum of heat colors indicates different migration weights in the migration event. Horizontal branch lengths are proportional to the amount of genetic drift occurring on the branch. The scale bar shows 10 times the average standard error of the entries in the sample covariance matrix. Refer to Figure 1 for abbreviations.

**Figure 3 animals-14-00699-f003:**
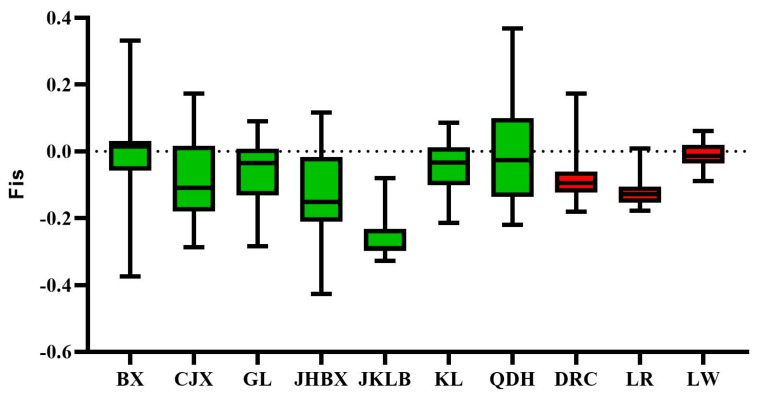
Genetic diversity of 7 pig breeds in Guizhou province: The histograms represent the inbreeding coefficients (Fis) of each breed. Pig breeds were plotted along the *X*-axis, while Fis values were plotted along the *Y*-axis. Green color represents local varieties in Guizhou Province and red color represents foreign varieties.

**Table 1 animals-14-00699-t001:** Genetic diversity of the 10 Eurasian pig breeds in this study.

Breed	Abbreviation	Number of Individuals	*Ho*	*He*	*Maf*	*F* _ROH_	*Fis*
Baixi	BX	30	0.31	0.31	0.23	0.019	−0.02
Congjiang Xiang	CJX	25	0.23	0.21	0.15	0.015	−0.09
Duroc	DRC	30	0.31	0.29	0.22	0.023	−0.07
Guanling	GL	30	0.28	0.27	0.19	0.009	−0.06
Jianhe White Xiang	JHBX	40	0.15	0.13	0.10	0.038	−0.12
Jiangkou Luobo	JKLB	16	0.39	0.31	0.24	0.009	−0.26
Kele	KL	22	0.36	0.35	0.26	0.008	−0.05
Landrace	LR	30	0.36	0.32	0.25	0.016	−0.12
Large White	LW	30	0.34	0.34	0.26	0.021	−0.01
Qiandong Hua	QDH	30	0.21	0.21	0.15	0.016	−0.01

**Table 2 animals-14-00699-t002:** Genetic differentiation index (*Fst*) of 10 Eurasian pig breeds in this study.

	CJX	DRC	GL	JHBX	JKLB	KL	LR	LW	QDH
BX	0.19	0.32	0.14	0.31	0.17	0.10	0.32	0.31	0.20
CJX		0.47	0.13	0.29	0.27	0.16	0.44	0.42	0.13
DRC			0.41	0.58	0.30	0.29	0.30	0.29	0.48
GL				0.25	0.21	0.10	0.38	0.35	0.12
JHBX					0.41	0.29	0.54	0.53	0.25
JKLB						0.14	0.31	0.29	0.28
KL							0.27	0.26	0.16
LR								0.24	0.45
LW									0.43

Refer to Table 1 for abbreviations.

## Data Availability

The data presented in this study are available upon request from the corresponding author.

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
