# Peer review of "Unveiling the Genetic Secrets of Chinese Indigenous Pigs from Guizhou Province: Diversity, Evolution and Candidate Genes Affecting Pig Coat Color"

_animals, 2024, doi:10.3390/ani14050699_

Round 1
Reviewer 1 Report
Comments and Suggestions for Authors
In this manuscript the authors used genome-wide single nucleotide polymorphism (SNP) markers to investigate the genetic diversity, historical correlations, and population structure of seven local pig breeds. Moreover, the authors identified a set of candidate genes that may have an impact on phenotypic differences in coat color in pigs.
The article has few minor deficiencies:
L4: ”and” must be placed before the last author.
L16: Please modify or clarify the word exobreeding.
L93-94: The authors describe here MAF, minor allele frequency?
L94-96: This phrase must be reformulated; the word team does not fit in the context.
L104: I ask the authors to mention which breed they are talking about because it is not clear.
L112-121. These sentences do not fall under the material and method section.
L136: minimum allele (Maf) probably is minor allele frequency (MAF)!?
L139: Each element in the formula must be described.
L141-142 ROH-Runs of homozygosity must be specified when it first appears in the text.
L160-163: The sentence must be revised.
Table 1: The abbreviations for table 1 are not mentioned.
L272-273: These results (all genes) can be published in the form of additional files. Moreover, I believe that the genes name should be in italic. Please be consistent throughout the manuscript.
I recommend the manuscript to be considered for publication after minor revision.
Reviewer 2 Report
Comments and Suggestions for Authors
This manuscript studied the genetic diversity, population structure, and evolutionary history of pig breeds in Guizhou Province, China. The research is comprehensive, covering various genetic aspects, including population structure, genetic diversity, and selection characteristics. The manuscript provides detailed insights into the genetic diversity among the pig breeds and their differentiation from European breeds, which is valuable for conservation efforts. It effectively identifies candidate genes that might influence traits like hair color, which is significant for further genetic improvement and understanding of phenotypic variation.
However, the paper might benefit from a larger and more diverse sample size to further validate the findings across different populations. While the research is thorough, it could emphasize more on the practical implications of these findings in pig breeding and conservation strategies. More comparative analysis with other regions' pig breeds could provide a broader context to the findings.
The Title: change to “Unveiling the Genetic Secrets of Chinese Indigenous Pigs from Guizhou Province: Diversity, Evolution and Candidate Genes Affecting Pig Hair Color”. “Historical Traits and Conservation Status” are not the main contents of this article.
Line 51: "wide distribution of local pig breeds, forming many excellent local pig breeds." - Repetition of "local pig breeds" could be avoided for better readability.
Line 53: “Erhualian pigs have the largest litter size in the world”, This lacks a reference. It’s “Erhualian”, not “Erhullian”. At the same time, it should be considered that the litter size of western pigs has also been greatly improved in the past 20 years. Whether Chinese local pigs (e.g. Erhualian) still have advantages in litter size needs the latest literature support.
Line 59-60: Moreover, the litter size of western pig breeds (e.g. Landrace from Denmark and Large White from France.) exceeds that of most local pigs in China now. This is one of the important reasons why western pig breeds occupy an absolute dominant market position in China.
Line 61: “Many unscientific hybridization methods introduced by China”, Not specific. Does this mean "Rotational crossbreeding"? Or something else? What does this have to do with the conservation of local pig breeds?
Line 139: The variables in the formula are not annotated.
Figure 2: Abbreviations lack annotation.
Table 1: Abbreviations lack annotation. The FROH and FIS results are inconsistent in JHBX pigs? How to explain it?
Table 2: What’s Fst? The interpretation of this abbreviation is inconsistent. Its calculation method is missing in Materials and Methods.
Line 260-280: “3.3. Screening of candidate genes affecting pig hair color” This part of the content suddenly appeared. Why analyze hair color? how to analyze candidate genes affecting hair color? and what are the hair color of these pig breeds? These should be explained in the Introduction or the Materials and Methods parts. The structure of the article needs readjustment.
Overall, the manuscript offers valuable insights into the genetic makeup of Chinese indigenous pigs. However, addressing the mentioned weaknesses could enhance the clarity and impact of the study.
Round 2
Reviewer 2 Report
Comments and Suggestions for Authors
The authors addressed my concerns. Thanks!